# Composting of Wild Boar Carcasses in Lithuania Leads to Inactivation of African Swine Fever Virus in Wintertime

**DOI:** 10.3390/pathogens12020285

**Published:** 2023-02-09

**Authors:** Tessa Carrau, Alvydas Malakauskas, Marius Masiulis, Paulius Bušauskas, Sigitas Japertas, Sandra Blome, Paul Deutschmann, Virginia Friedrichs, Simona Pileviečienė, Klaas Dietze, Daniel Beltrán-Alcrudo, Márk Hóvári, Gary A. Flory

**Affiliations:** 1Institute of Diagnostic Virology, Friedrich-Loeffler-Institut, Suedufer 10, 17493 Greifswald, Germany; 2Department of Veterinary Pathobiology, Veterinary Academy, Lithuanian University of Health Sciences, Tilzes 18, LT47181 Kaunas, Lithuania; 3Emergency Response Division, State Food and Veterinary Service, Siesiku 19, LT07170 Vilnius, Lithuania; 4Dr. L. Kriauceliunas Small Animal Clinic, Veterinary Academy, Lithuanian University of Health Sciences, Tilzes 18, LT47181 Kaunas, Lithuania; 5Center for Practical Training and Experimentation, Lithuanian University of Health Sciences, Akaciju 2, LT54310 Kaunas, Lithuania; 6Molecular Biology and GMO Department, National Food and Veterinary Risk Assessment Institute, J. Kairiukscio 10, LT08409 Vilnius, Lithuania; 7Institute of International Animal Health/One Health, Friedrich-Loeffler-Institut, Suedufer 10, 17493 Greifswald, Germany; 8Food and Agriculture Organization (FAO), Regional Office for Europe and Central Asia, 1054 Budapest, Hungary; 9G.A. Flory Consulting, Mount Crawford, VA 22801, USA

**Keywords:** African swine fever, African swine fever virus, wild boar, carcass disposal, composting, virus inactivation

## Abstract

African swine fever (ASF) continues to spread and persist in the Eurasian wild boar population. The infection pressure resulting from infected carcasses in the environment can be a major contributor to disease persistence and spread. For this reason, it is crucial to find a safe and efficient method of carcass disposal under different circumstances. In the presented study, we investigated open-air composting of carcasses under winter conditions in northeastern Europe, i.e., Lithuania. We can demonstrate that the ASF virus (ASFV) is inactivated in both entire wild boar carcasses and pieces thereof in a time- and temperature-dependent manner. Composting piles reached up to 59.0 °C, and ASFV was shown to be inactivated. However, the ASFV genome was still present until the end of the 112-day sampling period. While further studies are needed to explore potential risk factors (and their mitigation), such as destruction of composting piles by scavengers or harsh weather conditions, composting seems to present a valid method to inactivate the ASFV in wild boar carcasses where rendering or other disposal methods are not feasible. In summary, composting provides a new tool in our toolbox of ASF control in wild boar and can be considered for carcass disposal.

## 1. Introduction

African swine fever (ASF) is an infectious viral disease of domestic and wild pigs caused by a virus belonging to the *Asfaviridae* family, genus *Asfivirus*, with high lethality rates independently of age and sex leading to serious economic and production losses. Genotype II is responsible for the current panzootic that started in Georgia in 2007 and is still actively spreading throughout Eurasia and the Caribbean. The African swine fever virus (ASFV) can be sustained by wild boar populations for long periods without any involvement of domestic pigs, in what is considered a new transmission cycle not previously described [1,2].

African swine fever was introduced into Lithuania in 2014 [3,4]. Since then, the disease has reached up to 86% prevalence in the Lithuanian wild boar population by the end of 2018 [5]. Given that ASF is a highly lethal disease, wild boar carcasses are to be expected in affected populations. Such carcasses constitute a source of virus transmission for wild boar or free-ranging/scavenging pigs. Therefore, early detection through well-established passive surveillance and timely, safe disposal is of utmost importance [6]. Along these lines, the search for safe wild boar carcass disposal must be an integral part of the ASF control and eradication measures in the affected regions. Ideally, the disposal of recovered carcasses should be carried out at an animal by-products processing plant. However, this option does not always exist, as such facilities are not always present in some countries. This, combined with unreachable locations and/or exhausted facilities’ capacities, calls for other disposal options. Under these conditions, safe disposal must be carried out on site. In the European Union, rendering, burning, or burial are considered safe methods for carcass disposal (rules for animal by-products not intended for human consumption, Regulation (EC) 1069/2009 [7]), but flexibility is provided if safe disposal can be ensured. While burning is often rejected due to environmental concerns, burial has already been implemented in some countries, e.g., in Lithuania [8]. Burial, however, can be constrained by the climate in winter (frost), the structure of the habitat not allowing for adequately sized pits, e.g., in rocky terrains, a high water table, or the potential contamination of groundkwater when increased numbers of carcasses need to be disposed of. Under those circumstances, alternative methods such as composting should be explored. Composting as an “above-ground” disposal alternative was investigated for its effect on different swine pathogens by Pepin et al. [9]. Authors showed that the viability of common viruses such as porcine reproductive and respiratory syndrome virus (PRRSV) and porcine epidemic diarrhoea virus (PEDV) can be safely reduced by composting euthanized swine. Hence, carcass composting could present a safe and feasible option for mass disposal of infected carcasses, as in the context of ASF.

Traditionally, carcass composting involves covering carcasses with a carbon source [10,11,12]. Few studies focus on the evaluation of carcass composting in open-air outdoor settings. Nonetheless, Pepin et al. showed that composting pre-processed carcasses (carcasses ground into small particle sizes through mechanical crushing) achieved adequate temperatures for pathogen elimination. Additionally, Duc et al. [13] showed that composting swine in warm climates could effectively inactivate the ASF virus (ASFV).

The presented study aimed to analyze ASFV stability and inactivation after composting infectious wild boar carcasses under field and cold weather conditions. Process parameters such as composting temperatures were assessed. Furthermore, composting of entire carcasses and pre-processed materials were compared in terms of ASFV inactivation and eventual elimination.

## 2. Materials and Methods

### 2.1. Study Design

Two studies were conducted from November 2021 to February 2022. In both studies, intact wild boar carcasses were obtained by active search that was performed on a weekly basis in areas where increased ASF-related mortality had been detected in Lithuania. In total, nine ASFV-positive wild boar carcasses were chosen for the purpose of the studies (Figure 1A). Selection criteria were a good general state of preservation (i.e., estimated to be less than seven days old), no signs of scavenging, and no open body cavities. The carcass suitability assessment was completed visually by official veterinary inspectors of the corresponding Territorial Veterinary Service. The carcasses were subsequently transported following strict biosecurity protocols to a fenced field trial location at the Centre for Practical Training and Experimentation at Lithuanian University of Health Sciences in the Kaunas district municipality. For both studies, compost piles were composed by wheat straw piles in combination with sawdust. Studies 1 and 2 differed in the wild boar carcass preparation for sampling and in the sampling protocol.

#### 2.1.1. Study 1

On 4 November 2021, for study 1, *n* = 3 wild boar carcasses were obtained from a hunting ground in the Kretinga district municipality in the western part of Lithuania, where mass mortality of wild boar due to ASF had occurred for several months. Carcasses 1 and 2 belonged to young animals (less than 12 months of age; approx. 50 kg) and carcass 3 to an adult sow (over 24 months of age; approx. 100 kg) as shown in Figure 1A. For the purpose of this study, all carcasses were laid uncut on the compost pile, with the exception of two superficial incisions aimed to sample the inguinal lymph nodes at composting day 1. On the same day, blood and oro-nasal swabs were collected, and ASFV positivity was confirmed via PCR.

Once compost piles were established, sampling was conducted within 7-day intervals as follows:For carcass no. 1, samples were obtained on days 7, 14, 21, and 28 post-composting.For carcass no. 2, samples were obtained on days 14, 21, 28, and 35 post-composting.For carcass no. 3, samples were obtained on days 21, 28, 35, and 42 post-composting.

On each sampling day, the following matrices were collected: bone marrow, kidney, spleen, abdominal fluid, and sawdust from underneath the carcass, as listed in Appendix A. At each sampling, only part of the spleen and kidney were taken. The sampling was performed aiming for minimal carcass and pile disturbance. In the event of sawdust and/or straw removal, layers were restored after the sampling was finished. On day 112 (February 24, 2022), sampling concluded for all three carcasses by sampling of bone marrow and remaining mass of muscle tissue for each carcass.

#### 2.1.2. Study 2

For the second study, *n* = 6 wild boar carcasses were used. With the exception of carcass 4, which was found in the Vilnius district municipality in the southeast of Lithuania, the remaining carcasses (5–9) were found in a radius of 70 m in the western part of Lithuania, in the Skuodas district municipality, where active ASFV circulation and mass wild boar mortality was registered over a few months. Carcass 4 belonged to an adult male (≈36 months of age; approx. 100 kg), carcass 5 belonged to an adult sow (≈42 months of age; approx. 140 kg), and carcasses 6–9 belonged to young animals (<12 months of age; approx. 50 kg). Carcass 4 was placed for composting on 23 November, while the remaining five carcasses were placed on 29 November.

In contrast to study 1, all six carcasses were cut (as shown in Figure 1B) after placement on the compost pile (day 1). To this end, limbs were cut at articulation level, separating the front and hind legs from the main body, and organs (spleen and kidney) were divided into seven equal pieces. In order to avoid pile disturbance and to ease the extraction of each sampled body part from the carcass, the limbs (meant to sample the bone marrow) were tied to a cord, and the organs were wrapped with a gauze and, subsequently, also tied to a cord. The limbs, wrapped pieces of spleen, and kidneys were placed into each carcass as anatomically close as possible. Cords were marked and placed on the tops of the piles. ASFV positivity was confirmed via PCR from the bone marrow fluid, spleen, kidney, and fluid from abdominal cavity samples.

Once the compost pile was established, sampling was conducted on the following days: 1, 3, 8, 15, 22, and 29. The study concluded on February 24th (day 92 for carcass no. 4 and day 87 for carcasses 5–9) when the bones and the remaining mass of muscle tissue were sampled from all carcasses.

### 2.2. Compost Piles

Both studies were conducted at the same location in a fenced area of approximately 64 m^2^. The materials for composting used in both studies comprised a base layer of straw of at least 60 cm covered with approximately 20 cm of sawdust, on which the cadavers were placed. The piles were then covered with approximately 40 cm of straw and 20 cm of sawdust in order to obtain an absorptive layer (as shown in Figure 1A,C,D). The base of each compost pile was designed to be long and wide enough to accommodate the number of carcasses aimed to compost (approximately 45 cm from the edge of the base in all directions), which varied depending on the performed study.

#### Weather Monitoring and Composting Temperature

During the study, the daily weather condition monitoring data at the locations where the wild boar carcasses were found and the composting site were obtained from the closest weather monitoring stations (WMS) of the Lithuanian Hydrometeorological Service (www.meteo.lt/eng accessed on 27 July 2022). The data included the highest, lowest, and average temperatures, as well as the precipitation (rainfall and/or snowfall).

The data from the locations where the carcasses were found were collected retrospectively at least 7 days before the carcass detection. For study 1 the closest WMS was situated approx. 8.5 km from the acquired wild boar carcasses. For study 2, the data were collected approx. 22 km from carcass 4 and approx. 23 km from carcasses 5 to 9. Finally, the weather conditions from the compositing site were obtained from a WMS situated at approx. 6.5 km distance.

Next, the composting temperatures were monitored with calibrated 0.5 m long stainless steel compost thermometer probes (resolution 1 °C, accuracy ±1 °C, Electronic temperature instruments LTD). During each measurement, three probes were placed at the following locations in each pile: one in an area of a carcass back, one in the area between the front leg and head, and one in the area of the carcass rear site.

The temperatures were collected every other day at least three times per week, starting from the first day (5 November) after setting the composting piles until the last planned sampling (day 29 of study 2, 27 December). From that day onward, the temperatures were measured once a week until the end of the field trial on 24 February.

### 2.3. ASFV Infectivity

#### 2.3.1. Sample Analysis

All collected samples from the composting site were sent for polymerase chain reaction (PCR) examination at the National Food and Veterinary Risk Assessment Institute, which is the National Reference Laboratory (NRL) for ASF in Lithuania. The laboratory methods and sampling procedure were deployed as previously described [14].

At the end of the field trial, part of each sample was stored at −80 °C and subsequently sent to the German NRL, the Friedrich-Loeffler-Institut (FLI), for virus isolation.

#### 2.3.2. Virus Isolation

Following the PCR diagnosis at the Lithuanian NRL, the samples were tested at FLI for virus isolation to assess their infectivity. First, each sample was subjected to one blind passage in porcine peripheral blood monocytic cell (PBMC)-derived macrophages, followed by a read-out passage, as previously described [15]. For the blind passages, 5 × 10^6^ PBMCs per well were seeded into 24-well Primaria plates (Corning, Durham, NC, USA) two days before inoculation. After 2 h incubation at 37 °C to allow virus adsorption, the cells were washed once with PBS, and fresh medium was added. The cultures were examined daily with a light microscope and were cultivated for 5 days at 37 °C. After a freeze–thaw cycle, virus titrations were carried out as previously described [16].

For proof of infectious virus, 7.5 × 10^4^ PBMCs per well were seeded in a 96-well Primaria tissue culture plate (Corning). The following day, the attached cells were inoculated with 100 µL of supernatant from the blind or second passage in tenfold dilutions from 10^−1^ to 10^−8^. One day after inoculation, 20 µL of a 1% solution of homologous red blood cells were added to each well. The plates were examined after 72 h. Each well with at least one hemadsorbing macrophage read under a microscope, indicative of virus replication, was considered positive. The titers were calculated by the Spearman–Kärber method and expressed as log_10_ HAD_50_, with a limit of detection of 10^1.75^ per mL.

## 3. Results

### 3.1. Temperature Monitoring

#### 3.1.1. Weather Conditions at Locations Where Carcasses Were Found

The timeframe covered the estimated period from the wild boar’s death to the carcass recovery. This aimed to record the possible environmental influence on the composted carcasses. The recorded data are listed in Appendix A. The carcasses used in study 1 originated from the Kretinga district municipality, in the northwest part of Lithuania, where the mean temperature was 8.4 ± 2.5 °C and the mean precipitation was 5.2 ± 7.1 mm during the last 7 days before the start of the study. The carcasses used in study 2 originated from the Vilnius district municipality in the southeast of Lithuania and the Skuodas district municipality in the northwest part of Lithuania, where the mean temperatures were 3.8 ± 3.2 °C and 3.7 ± 3.0 °C, respectively. The mean precipitation in the Vilnius district municipality during that timeframe was 2.3 ± 4 mm, while in the Skuodas district municipality, the mean precipitation was 0.9 ± 3.1 mm.

#### 3.1.2. Weather Conditions at Composting Site

The mean daily temperature at the composting site for the 112-day composting period (4 November 2021 to 28 February 2022) was 0.5 °C, with a maximum value of 9.1 °C on 5 November and minimum of −11.4 °C on 8 December (Figure 2). The precipitation and snowfall (cm) at the composting site were also monitored during the same period. An average of 2 mm precipitation was observed across 79 days during the study, and snowfall was observed for periods ranging from 2 (during November and February) to 23 days (during December and January) with up to 10 cm of snow coverage, as described in Figure 2.

#### 3.1.3. Recorded Temperatures in the Composting Piles

In study 1, a slow increase in windrow temperatures was recorded in November, with maximum temperatures reaching 21 °C. From December onward, the mean temperature steadily increased, with maximum temperatures reaching or exceeding 40.0 °C. The highest temperature was 47.0 °C in January 2022. However, high variability was observed while the mean and minimum steadily increased (see Figure 3A).

Overall, the piles in study 2 reached even higher maximum temperatures but needed more time to heat (see Figure 3B). In November, these piles did not exceed 8.4 °C. However, in December, a maximum temperature of 42.0 °C was recorded. In January and February, the windrow temperatures reached almost 60 °C, with a maximum of 59 °C in January 2022 and 54.0 °C in February 2022. Again, variability was high, with steadily increasing mean and minimum temperatures.

It is also noteworthy that high pile temperatures were reached despite the cold air temperatures. For example, a mean air temperature of −0.1 ± 2.9 °C corresponded to mean pile temperatures above 35 °C in study 2. Under snow coverage (e.g., 12 cm on 1 December, post-windrow-formation on day 13 for study 1) the recorded temperatures inside the pile showed an average of 19.7 ± 1.4 °C.

### 3.2. ASFV Infectivity and DNA Analysis in Collected Samples

To determine whether composting both entire and dissected wild boar carcasses abolished ASFV infectivity, regularly collected samples (as described in the materials and methods Section 2.3) were passaged in PBMC-derived macrophages. At day 1 post-windrow-formation (pwf), both composting piles tested positive for ASFV (Figure 4). However, the virus infectivity decreased over the course of the study. The earliest time pwf with a lack of detectable infectivity was identified at days 42 and 15 for studies 1 and 2, respectively.

Positive PCR results for the ASFV genome were recorded until days 112 and 92 of studies 1 and 2, respectively (Figure 4). The results were organ-related, and all PCR Ct values are listed in Appendix A. With the exception of carcass 8, ASFV DNA was detected in the bone marrow and muscle tissue of all composted carcasses until the last sampling (day 112). The blood tested positive in carcass 1 and in the organ of carcass number 4. The organ samples and abdominal fluid tested negative by PCR after sampling day 35.

## 4. Discussion

African swine fever continues to spread in the Eurasian wild pig population, and the infection pressure resulting from carcasses in the environment can be a considerable contributor to disease spread and habitat-bound persistence [17]. For this reason, it is crucial to find a safe and efficient method of wild boar carcass disposal under different environmental (habitats, soil compositions, and materials) and weather conditions. In terms of process safety and economy, rendering is demonstrated to be the most effective way to dispose of carcasses. However, not all countries have their own rendering plants, and some plants may not accept wild animal carcasses. Moreover, the transportation of carcasses can pose transmission risks, and in some areas, removal and rendering might not be feasible, e.g., in poor-transport-infrastructure regions. In these cases, alternatives such as burial, burning, and composting could aid the implemented measures. The latter has not been systematically assessed under field conditions for wildlife carcasses in general nor for ASFV-infected wild boar.

The main objective of this study was to analyze the potential inactivation of ASFV in naturally infected wild boar carcasses through composting under open-air and cold weather conditions. In this study, researchers compared two different means of wild boar carcass processing for composting, analyzing each for their potential to achieve adequate conditions for ASFV elimination. In both scenarios, inactivation of ASFV was achieved over time even under cold weather conditions. Taking the temperature in the pile into consideration, this is in line with expectations built on virus half-life and decimal reduction rates presented by Mazur-Panasiuk and Wozniakowki in 2020 [18]. The estimated half-life of the virus at 23 °C was 0.38 days. Since much higher maximum temperatures were achieved even in winter, composting was confirmed as a method capable of virus inactivation. This made clear that additional factors beyond temperature play a role in inactivation. This was also seen in biogas plants, where bacteria and their metabolism products were reported to positively impact inactivation times [19]. The difference in inactivation times between studies 1 and 2 can be explained by different maximum temperatures. In the study 1 compost pile, the temperatures reached >45 °C for at least three consecutive days corresponding to longer infectivity of the composted carcasses than in study 2. The potential infectivity of the compost material the first weeks after windrow formation is a known phenomenon described by previous authors [9]. Additionally, when intact swine carcasses are composted, it has been estimated to require at least 40 hrs of exposure at 60 °C for the carcass center to reach 56 °C [20]. The bone marrow of ASF-infected pigs is known to contain a high viral load. Therefore, reaching high internal temperatures is critical [21]. Nonetheless, a limitation of this study was that the compost temperatures were not recorded frequently enough from January onward to confirm the potential increase in the pile temperature, which might indeed have reached higher temperatures. It should be noted that the carcasses in study 2 approached temperatures as high as 59 °C. In previously related studies, it was concluded that after reaching 60 °C, virus inactivation occurred in 15 to 20 min [18]. In addition to infectivity reduction, an impact was observed on the detectability of ASFV genome, demonstrating strong degradation of all matrices in the pile [18].

Previous studies stressed the importance of placing biomass underneath the carcass compost material as it might prevent the contamination of groundwater with viral residues [9]. To mitigate this potential risk, the composting piles were composed of a thick layer (~60 cm) of straw. Under study settings, frozen soil under the piles should have prevented the pathogen leakage further, as was demonstrated by Gupta et al. [22] and Jamieson et al. [23]. However, further research is needed to assess the potential leaking of AFSV to the groundwater from contaminated carcasses, particularly during warmer weather.

Under field conditions, there is also the risk of pile destruction by scavengers and predators. The piles for the two studies presented here were fenced to mitigate this risk, and no destructive interference was observed. It could be advisable to fence all potential composting areas and to monitor them with game cameras when composting as carcass disposal within disease control interventions. Biosecurity measures must also be strictly followed during the composting process, including the use of appropriate prevention methods during hunting and carcass transportation [17,24].

This study demonstrated that composting piles could reach adequate temperatures and reduce/abolish ASFV infectivity even in cold climate and winter conditions. Similar results had previously been obtained for PRRSV by Pepin et al. [9], who evaluated similar parameters under natural conditions that occur during outdoor winter weather situations. This study determined that unprocessed carcasses can also be composted with a significant virus infectivity reduction. To date, composting studies have been carried out with pre-processed carcasses, successfully eliminating viruses such as foot and mouth disease (FMD), PEDV, or PRRSV [10,25,26]. The prolonged detection of ASFV in the collected samples is likely due to the high stability of double-stranded DNA viruses (Grosjean, 2009).

## 5. Conclusions

European Union legislation does not provide specific rules for dealing with the safe disposal of carcasses found in affected regions. This leaves the competent authority room for the implementation of these measures. In the FAO Animal Production and Health Manual on African swine fever in wild boar [17], different strategies are mentioned, i.e., incineration, rendering, and on-the-spot burial or burning. While rendering or incineration are the most effective and reliable ways to dispose of carcasses, safe transportation to the rendering sites can be hampered by inaccessible habitat, and many countries lack suitable rendering plants. Under these circumstances, disposal methods that can be carried out on the spot, or at least locally, become an important option. Among these are burial (single pit, on-site trench, or mass burial) and burning, and both are mentioned in the respective manual [17]. However, as determined by this research, composting is also a valid option. Burning can cause environmental pollution and might not be feasible in the dry season with a high risk of forest fires. Burial, on the other hand, is difficult to perform in the winter season when digging is hampered by the frozen ground. Composting demonstrated to be effective in wintertime can be an important addition to the toolbox of carcass disposal, especially for countries that lack rendering capacity. As such, this technique should be included in any new edition of the FAO manual providing guidance for affected countries inside and outside the EU.

## Figures and Tables

**Figure 1 pathogens-12-00285-f001:**
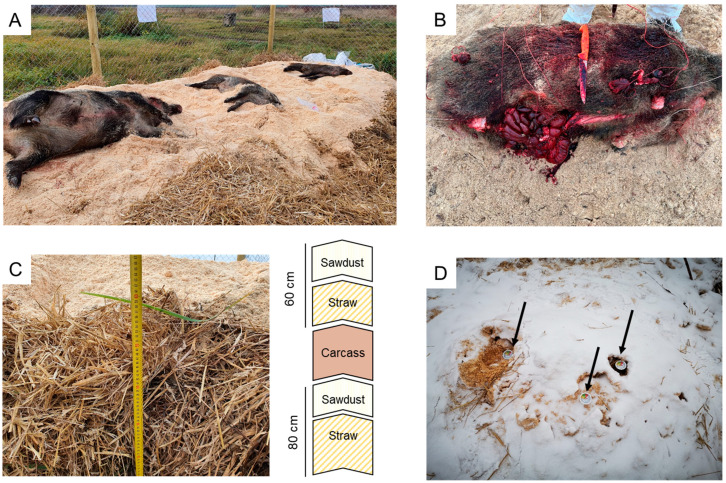
Composting piles for ASFV-infected wild boar carcasses. Two different approaches were carried out: in the first (**A**), uncut carcasses were placed on the sawdust, while in the second approach (**B**), carcasses were cut for later sampling before finishing the formation of a pile. In both studies, (**C**) straw of 60 cm high was placed in the base, and the carcasses were covered with approx. 40 cm of straw and 20 cm sawdust. (**D**) During the study, the temperature in the pile was regularly monitored using stainless steel compost thermometer probes (arrows).

**Figure 2 pathogens-12-00285-f002:**
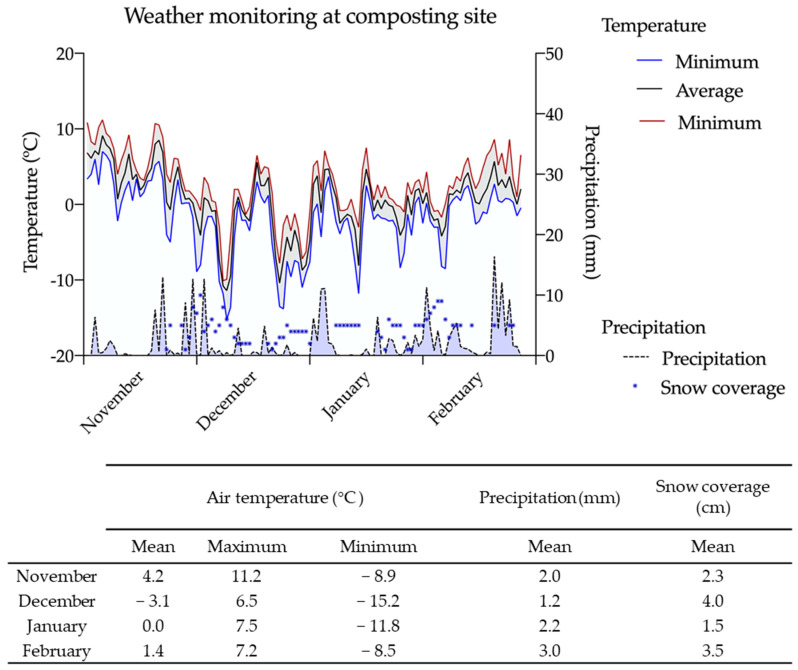
Weather conditions and temperature readings of the compost site.

**Figure 3 pathogens-12-00285-f003:**
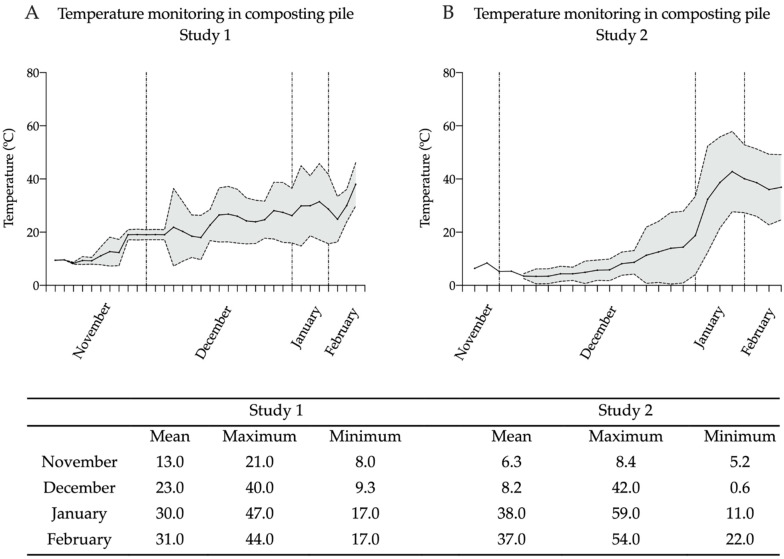
Temperature recordings at the compost piles.

**Figure 4 pathogens-12-00285-f004:**
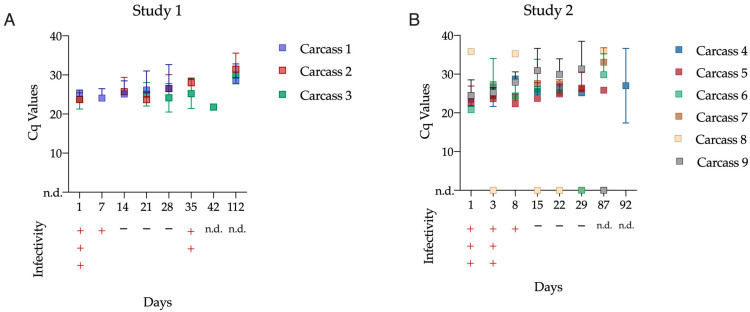
PCR and virus infectivity findings during the composting period. Squares represent the mean Ct value, and error bars depict the standard deviation per carcass. In study 1 (**A**), samples were found to be PCR positive until day 112. However, infectivity was found until day 35. In contrast, in study 2 (**B**), virus inactivation was shown to be faster, and no infectivity was found beyond day 8. In this latest study, PCR-positive samples were recorded until day 112. N.d.: not determined.

## Data Availability

Data are available on request from the corresponding author.

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
