# Peer review of "Composting of Wild Boar Carcasses in Lithuania Leads to Inactivation of African Swine Fever Virus in Wintertime"

_pathogens, 2023, doi:10.3390/pathogens12020285_

Round 1

Reviewer 1 Report

Presentation of results is clearly laid out and in proper sequence. Graphical illustration of results  is properly designed with self explanatory. The role of other anaerobic bacteria and generated fluid in inactivation of ASF virus could have analysed. 

Author Response

Reviewer 1

Presentation of results is clearly laid out and in proper sequence. Graphical illustration of results is properly designed with self-explanatory. The role of other anaerobic bacteria and generated fluid in inactivation of ASF virus could have analysed.

Authors thank the reviewer for the comment, the role of anaerobic bacteria was beyond the scope of this project but we acknowledge the importance of the topic for a future works.

Reviewer 2 Report

In my opinion the paper very good written, easy to read and the study by itself useful and interesting for veterinarians and hunters. I have no comments on research but want to point on that the potential readers can outside from EU, include hunters and they can directly accept this protocol. Of course there are some reference in the paper to EU reconditions for carcasses disposal but it can be useful to point additionally that minimal biosecurity measures require to manage infected carcasses including PPP using and fomites disinfection. It is not clear from the text and can mislead readers.

Author Response

In my opinion the paper very good written, easy to read and the study by itself useful and interesting for veterinarians and hunters. I have no comments on research but want to point on that the potential readers can outside from EU, include hunters and they can directly accept this protocol. Of course, there are some reference in the paper to EU reconditions for carcasses disposal but it can be useful to point additionally that minimal biosecurity measures require to manage infected carcasses including PPP using and fomites disinfection. It is not clear from the text and can mislead readers.

Authors thank the reviewer for the suggestion. We have added the following statement in L340 “Biosecurity measures must also be strictly followed during the composting process, this includes the use of appropriate preventive methods during hunting and further carcass transportation [16, 23]” and linked to the paper Preventive measures aimed at minimizing the risk of African swine fever virus spread in pig farming systems which includes a chapter named “Preventive measures during hunting”.

Reviewer 3 Report

African swine fever is a serious threat to the global pig industry development. The infection pressure resulting from infected carcasses in the environment can be a major contributor to disease persistence and spread. The main goal of this study was to analyze the potential inactivation of ASFV in naturally infected wild boar carcasses through composting under open-air and cold weather conditions. In this study, two different ways of wild boar carcass processing for composting and analyzed were compared, each of them for their potential to achieve adequate conditions for ASFV elimination. Results demonstrated that ASF virus (ASFV) was inactivated in both entire wild boar carcasses and pieces thereof in a time- and temperature-dependent manner.

This study provides us with a safe, clean and low-carbon method for disposing ASFV infected pig carcasses, especially for countries that lack rendering capacity.

In general, results are convincingly presented and constitute an interesting contribution to the disposal method for ASFV infected wild boar carcasses.

Minor concerns:

1.     In Fig.2 and Fig.3, table and figures were mixed together, figures and tables should be presented separately.

2.     English language should be polished throughout the manuscript.

Author Response

Reviewer 3

African swine fever is a serious threat to the global pig industry development. The infection pressure resulting from infected carcasses in the environment can be a major contributor to disease persistence and spread. The main goal of this study was to analyze the potential inactivation of ASFV in naturally infected wild boar carcasses through composting under open-air and cold weather conditions. In this study, two different ways of wild boar carcass processing for composting and analyzed were compared, each of them for their potential to achieve adequate conditions for ASFV elimination. Results demonstrated that ASF virus (ASFV) was inactivated in both entire wild boar carcasses and pieces thereof in a time- and temperature-dependent manner.

This study provides us with a safe, clean and low-carbon method for disposing ASFV infected pig carcasses, especially for countries that lack rendering capacity.

In general, results are convincingly presented and constitute an interesting contribution to the disposal method for ASFV infected wild boar carcasses.

Minor concerns:

  1. In Fig.2 and Fig.3, table and figures were mixed together, figures and tables should be presented separately.

We thank the reviewer for the suggestion but we believe that this allows a better understanding of the presented results since figures and tables are linked together.

  1. English language should be polished throughout the manuscript.

Authors will proof read the written English before resubmitting the manuscript.

Reviewer 4 Report

The authors of the present study evaluated the suitability of open-air composting during wintertime for the disposal of carcasses of African swine fever virus (ASFV)-infected wild boar. For this purpose, they collected ASFV-positive wild boar carcasses from the field and evaluated two procedures, either with unopened carcasses (n=3) or with cut carcasses (n=6). The studies were conducted for a duration of 4 months. The following parameters were monitored at regular intervals: weather conditions (temperatures and precipitations) at the finding locations of the carcasses, weather conditions at the composting site, temperatures in the composting piles, infections ASFV in the carcasses by isolation in cell culture, ASFV genome by qPCR. They report that no infectious virus could be recovered from any carcasses on any sampling time beyond 35 days of composting. Virus inactivation was faster when the carcasses were cut. Nevertheless, viral DNA was detected until the end of the study, for 112 days. These data show that composting is a suitable method for the disposal of ASFV-positive carcasses, even at low ambient temperatures such as in wintertime.

General comment: This is an interesting descriptive study that shows that ASFV can be completely inactivated by composting of infected carcasses. It is certainly not the first-choice procedure for disposal of carcasses, but the data is worth being reported since it can be of use under certain circumstances. Nevertheless, there are a few points that should be addressed or clarified before the manuscript can be published.

Comment 1: The sentence of line 50 is not clear. What does 86% mean? Were 86% of the wild boars infected? The grammar of this sentence must be revised.

Comment 2: On line 104, correct 2022 with November 4th 2021!

Comment 3: What do the arrows represent in figure 1D? This must be mentioned in the legend of figure 1 at least.

Comment 4: the ASFV isolation in monocyte-derived macrophages (lines 196 to 206) is not described properly. Where the monocytes enriched or purified from the PBMC fraction and how? How were the monocytes differentiated to macrophages, what medium, serum and factors were used? Please describe accurately and provide appropriate references.

Comment 5: in the graph of figure 2, since temperature and precipitation are represented in the same graph, Minimum, Average, Maximal (use Maximum) should refer explicitly to the temperature (although it is obvious).

Comment 6: On line 257, after “regularly”, refer to materials and methods in brackets for instance

Comment 7: on lines 260 to 263, the authors state: “In the first scenario (Study 1) samples obtained at 7 and 35 days pwf were still infectious, but samples obtained from that day onward showed lack of infectivity. A slightly different scenario was observed for the second study, where no virus infectivity was displayed from day 8 pwf onwards.” These statements are a bit misleading because the earliest sampling date with lack of detectable infectivity following the day indicated in the text was 7 days later, i.e. day 42 and 15, respectively. Therefore, please indicate the earliest time pwf for which all samples were virus negative.

Comment 8: related to comment 7, labeling of the x-axes of graphs 4A and B is incomplete. What is the meaning of blanks with respect to infectivity? Please fill all spots, either with “+”, “-“, or “n.d.” not determined (or whatever the blank means).

Comment 9: The sentence of lines 264-265 is misleading and in contradiction with the following lines 266-270, figure 4 and table S2. It says that “Positive PCR results for ASFV genome were recorded until day 42 of the study as shown in Figure 4”, while figure 4 and Table S2 show positive PCR results beyond that day, until day 112. Please revise!

Comment 10: related to comment 9, it is not clear which PCR results are shown in panels 4A and B. The colors of the squares relate to the carcasses. According to the text and table S2, several samples per carcass were analyzed on each sampling time. What does the square represent? The mean of all samples for a defined carcass? What die the error bars represent? Please revise the figure legend accordingly.

Comment 11: on lines 297-298, what do the authors mean by “in the respective study”? Do they mean “in the latter study”? Please clarify/correct.

Comment 12: on lines 303 to 305, it is not clear what is compared with what. Write for instance “The difference in inactivation times between study 1 and 2” (line 303], and on line 305 “corresponded to a longer infectivity of the composted carcasses than in study 2.”

Comment 13: The meaning/argumentation of the next sentence is not clear (lines 305-307). Why “However”? Please revise.

Comment 14: The meaning of the sentence of lines 328-330 is unclear. Please reformulate and revise the grammar.

Comment 15: overall, there are numerous language errors.

Author Response

Reviewer 4

Comments and Suggestions for Authors

The authors of the present study evaluated the suitability of open-air composting during wintertime for the disposal of carcasses of African swine fever virus (ASFV)-infected wild boar. For this purpose, they collected ASFV-positive wild boar carcasses from the field and evaluated two procedures, either with unopened carcasses (n=3) or with cut carcasses (n=6). The studies were conducted for a duration of 4 months. The following parameters were monitored at regular intervals: weather conditions (temperatures and precipitations) at the finding locations of the carcasses, weather conditions at the composting site, temperatures in the composting piles, infections ASFV in the carcasses by isolation in cell culture, ASFV genome by qPCR. They report that no infectious virus could be recovered from any carcasses on any sampling time beyond 35 days of composting. Virus inactivation was faster when the carcasses were cut. Nevertheless, viral DNA was detected until the end of the study, for 112 days. These data show that composting is a suitable method for the disposal of ASFV-positive carcasses, even at low ambient temperatures such as in wintertime.

General comment: This is an interesting descriptive study that shows that ASFV can be completely inactivated by composting of infected carcasses. It is certainly not the first-choice procedure for disposal of carcasses, but the data is worth being reported since it can be of use under certain circumstances. Nevertheless, there are a few points that should be addressed or clarified before the manuscript can be published.

Comment 1: The sentence of line 50 is not clear. What does 86% mean? Were 86% of the wild boars infected? The grammar of this sentence must be revised.

We have rephrased the statement to point out that the 86% was meant to express the reached prevalence in Lithuania: “Since then, the disease reached up to 86% prevalence through the Lithuanian wild boar population by the end of 2018”.

Comment 2: On line 104, correct 2022 with November 4th 2021!

We thank the reviewer for the correction, we have addressed the correct date.

Comment 3: What do the arrows represent in figure 1D? This must be mentioned in the legend of figure 1 at least.

We have added the following statement in the Figure 1D: “(D) During the study, temperature in the pile was regularly monitored using stainless-steel compost thermometer probes (arrows).”

Comment 4: the ASFV isolation in monocyte-derived macrophages (lines 196 to 206) is not described properly. Where the monocytes enriched or purified from the PBMC fraction and how? How were the monocytes differentiated to macrophages, what medium, serum and factors were used? Please describe accurately and provide appropriate references.

Authors agree in the lack of information; therefore, we have addressed the point and added a relevant reference citing all the hereby used protocol (Friedrichs et al., 2023).

Comment 5: in the graph of figure 2, since temperature and precipitation are represented in the same graph, Minimum, Average, Maximal (use Maximum) should refer explicitly to the temperature (although it is obvious).

We have added the missing information to the figure 2.

Comment 6: On line 257, after “regularly”, refer to materials and methods in brackets for instance

The following reference was added in L 269: “regularly (as described in material and methods section 2.3)”

Comment 7: on lines 260 to 263, the authors state: “In the first scenario (Study 1) samples obtained at 7 and 35 days pwf were still infectious, but samples obtained from that day onward showed lack of infectivity. A slightly different scenario was observed for the second study, where no virus infectivity was displayed from day 8 pwf onwards.” These statements are a bit misleading because the earliest sampling date with lack of detectable infectivity following the day indicated in the text was 7 days later, i.e. day 42 and 15, respectively. Therefore, please indicate the earliest time pwf for which all samples were virus negative.

For a better understanding, we have rephrased the statement: “The earliest time pwf with lack of detectable infectivity was found at days 42 and 15 for studies 1 and 2, respectively”

Comment 8: related to comment 7, labeling of the x-axes of graphs 4A and B is incomplete. What is the meaning of blanks with respect to infectivity? Please fill all spots, either with “+”, “-“, or “n.d.” not determined (or whatever the blank means).

Figure has been changed and the abbreviation of “not determined” (n.d.) was added to the x-axe.

Comment 9: The sentence of lines 264-265 is misleading and in contradiction with the following lines 266-270, figure 4 and table S2. It says that “Positive PCR results for ASFV genome were recorded until day 42 of the study as shown in Figure 4”, while figure 4 and Table S2 show positive PCR results beyond that day, until day 112. Please revise!

We thank the reviewer for the comments, we have addressed the misleading statement.

Comment 10: related to comment 9, it is not clear which PCR results are shown in panels 4A and B. The colors of the squares relate to the carcasses. According to the text and table S2, several samples per carcass were analyzed on each sampling time. What does the square represent? The mean of all samples for a defined carcass? What die the error bars represent? Please revise the figure legend accordingly.

The following information has been added to the legend: “Squares represent the mean Ct value and error bars depict the standard deviation per carcass”

Comment 11: on lines 297-298, what do the authors mean by “in the respective study”? Do they mean “in the latter study”? Please clarify/correct.

The according change has been addressed.

Comment 12: on lines 303 to 305, it is not clear what is compared with what. Write for instance “The difference in inactivation times between study 1 and 2” (line 303], and on line 305 “corresponded to a longer infectivity of the composted carcasses than in study 2.”

We thank the reviewer and have corrected the statements.

Comment 13: The meaning/argumentation of the next sentence is not clear (lines 305-307). Why “However”? Please revise.

We have removed the argumentation.

Comment 14: The meaning of the sentence of lines 328-330 is unclear. Please reformulate and revise the grammar.

We have addressed the grammatical mistakes

Comment 15: overall, there are numerous language errors.

We have reviewed and provided English language corrections.